# Evaluation of Gastric pH and Gastrin Concentrations in Horses Subjected to General Inhalation Anesthesia in Dorsal Recumbency

**DOI:** 10.3390/ani14081183

**Published:** 2024-04-15

**Authors:** Jesus Leonardo Suarez Guerrero, Pedro Henrique Salles Brito, Marília Alves Ferreira, Julia de Assis Arantes, Elidiane Rusch, Brenda Valéria dos Santos Oliveira, Juan Velasco-Bolaños, Adriano Bonfim Carregaro, Renata Gebara Sampaio Dória

**Affiliations:** 1Department of Veterinary Medicine, School of Animal Science and Food Engineering, University of São Paulo (USP), 225 Duque de Caxias Norte Avenue, Pirassununga 13635-900, SP, Brazil; jesusuarezg@alumni.usp.br (J.L.S.G.); pedro.brito@usp.br (P.H.S.B.); marilia.alves.ferreira@usp.br (M.A.F.); julia.arantes@usp.br (J.d.A.A.); elidianerusch@usp.br (E.R.); brendamv@usp.br (B.V.d.S.O.); carregaro@usp.br (A.B.C.); 2Grupo de Investigación en Ciencias Agropecuarias (Group GIsCA), Facultad de Medicina Veterinaria y Zootecnia, Institución Universitaria Visión de las Américas, Pereira 660003, Colombia; juan.velasco@uam.edu.co; 3Research Group Calidad de Leche y Epidemiología Veterinária (CLEV), Universidad de Caldas, Manizales 170004, Colombia

**Keywords:** anesthesia, recumbency, gastrin, gastric ulcer, gastroenterology, horses

## Abstract

**Simple Summary:**

Horse stomach problems are common, with gastric ulcers being the most common. These diseases affect horses’ athletic performance and their health. This study evaluated the pH of the stomach of anesthetized horses in dorsal recumbency and also for 24 h after anesthesia, seeking to evaluate the harmful effects of anesthesia on the horse’s stomach. It was found that anesthesia and lying positioning did not affect the pH of the stomach, which remained acidic. After waking from anesthesia, the pH became slightly basic, as with the pH of the stomach of non-anesthetized horses, whether fed or not. It was concluded that there is no need for protective measures prior to an anesthetic procedure in horses.

**Abstract:**

The prevalence of gastric disorders in high-performance horses, especially gastric ulceration, ranges from 50 to 90%. These pathological conditions have negative impacts on athletic performance and health. This study was designed to evaluate changes in gastric pH during a 24 h period and to compare gastrin concentrations at different time points in horses undergoing general inhalation anesthesia and dorsal recumbency. Twenty-two mixed-breed mares weighing 400 ± 50 kg and aged 8 ± 2 years were used. Of these, eight were fasted for 8 h and submitted to 90 min of general inhalation anesthesia in dorsal recumbency. Gastric juice samples were collected prior to anesthesia (T0), and then at 15 min intervals during anesthesia (T15–T90). After recovery from anesthesia (45 ± 1 min), samples were collected every hour for 24 h (T1 to T24) for gastric juice pH measurement. During this period, mares had free access to Bermuda grass hay and water and were fed a commercial concentrate twice (T4 and T16). In a second group (control), four non-anesthetized mares were submitted to 8 h of fasting followed by nasogastric intubation. Gastric juice samples were then collected at T0, T15, T30, T45, T60, T75, and T90. During this period, mares did not receive food or water. After 45 min, mares had free access to Bermuda grass hay and water, and gastric juice samples were collected every hour for four hours (T1 to T4). In a third group comprising ten non-fasted, non-anesthetized mares with free access to Bermuda grass hay and water, gastric juice samples were collected 30 min after concentrate intake (T0). In anesthetized mares, blood gastrin levels were measured prior to anesthesia (8 h fasting; baseline), during recovery from anesthesia, and 4 months after the anesthetic procedure, 90 min after the morning meal. Mean values of gastric juice pH remained acidic during general anesthesia. Mean pH values were within the physiological range (4.52 ± 1.69) and did not differ significantly between time points (T15–T90; *p*  >  0.05). After recovery from anesthesia, mean gastric pH values increased and remained in the alkaline range throughout the 24 h period of evaluation. Significant differences were observed between T0 (4.88 ± 2.38), T5 (7.08 ± 0.89), T8 (7.43 ± 0.22), T9 (7.28 ± 0.36), T11 (7.26 ± 0.71), T13 (6.74 ± 0.90), and T17 (6.94 ± 1.04) (*p*  <  0.05). The mean gastric juice pH ranged from weakly acidic to neutral or weakly alkaline in all groups, regardless of food and water intake (i.e., in the fasted, non-fasted, and fed states). Mean gastric pH measured in the control group did not differ from values measured during the 24 h post-anesthesia period or in the non-fasted group. Gastrin concentrations increased significantly during the post-anesthetic period compared to baseline (20.15 ± 7.65 pg/mL and 15.15 ± 3.82 pg/mL respectively; *p*  <  0.05). General inhalation anesthesia and dorsal recumbency did not affect gastric juice pH, which remained acidic and within the physiological range. Gastric juice pH was weakly alkaline after recovery from anesthesia and in the fasted and fed states. Serum gastrin levels increased in response to general inhalation anesthesia in dorsal recumbency and were not influenced by fasting. Preventive pharmacological measures are not required in horses submitted to general anesthesia and dorsal recumbency.

## 1. Introduction

Equine gastric ulcer syndrome (EGUS) is a multifactorial condition with a reported prevalence ranging from 53 to 93% [1,2]. This syndrome has several causes, including stress, prolonged fasting, diet, athletic activity, underlying diseases, and drugs, among others [1,2]. Athletic horses are primarily affected due to chronic stress, intense training, and controlled feeding [3]. Clinical signs of EGUS range from mild to severe recurrent abdominal pain, behavioral changes, decreased performance, hypersalivation, bruxism, anorexia, and weight loss, and may eventually lead to death [1,4,5].

Gastroesophageal reflux disease (GERD) and duodenogastroesophageal reflux disease (DGER) are similar conditions described in humans. Increased intra-abdominal pressure resulting from obesity and recumbent position is thought to be a risk factor in affected patients [6,7,8]. Head-of-bed elevation with an elevation angle of 22° to prevent reflux while lying down is recommended in human patients with GERD and DGER [9]. In horses, general inhalation anesthesia and dorsal recumbency may predispose them to duodenogastric and gastroesophageal reflux, compromising the integrity of the gastric and esophageal mucosa and ultimately leading to gastric and esophageal ulceration and EGUS [10,11].

Continuous pH measurement has excellent sensitivity (77% to 100%) and specificity (88% to 100%) and is the diagnostic method of choice for GERD and DGER [6,8]. Using a pH meter placed in the distal esophagus, next to the cardia, gastric pH can be monitored continuously for 24 h. This method is the gold standard to determine how long the gastric juice remains in contact with the esophageal mucosa and to detect peak pH values, which can then be related to clinical signs and symptoms [12]. Alternatively, serum gastrin concentrations can be measured. Gastrin enhances hydrochloric acid secretion into the stomach and plays an important role in gastric physiology. Gastrin concentrations are routinely measured by radioimmunoassay to investigate variations during prolonged fasting [13,14].

Assessment of gastric pH during and after general anesthesia and dorsal recumbency has not been described in horses to date. This study was designed to monitor gastric juice pH for a period of 24 h in horses submitted to general inhalation anesthesia in dorsal recumbency. Changes in gastrin concentrations at different time points were also investigated.

## 2. Material and Methods

### 2.1. Animals

Twenty-two healthy, adult mixed-breed mares aged 8 ± 2 years and weighing 400 ± 50 kg were included in the study. Mares were considered clinically healthy based on findings of routine physical examination and hematological and biochemical profiles. Mares were housed indoors throughout the experimental period and fed a maintenance diet consisting of Bermuda grass (Cynodon dactylon) hay and water ad libitum and commercial horse feed.

### 2.2. Ethical Statement

The procedures used in this study were approved by the Ethics Committee on Animal Use of the School of Animal Science and Food Engineering of the University of São Paulo protocol n° 1773270819.

### 2.3. Experimental Groups

The mares in this study were allocated to one of three groups: the anesthetized group, control group, or non-fasted group (8, 4, and 10 mares, respectively). Mares in the anesthetized group were submitted to 8 h of fasting followed by nasogastric intubation, general inhalation anesthesia, dorsal recumbency, and gastric fluid sample collection. Mares in the control group were also fasted for 8 h and submitted to nasogastric intubation and sample collection but were not anesthetized. Mares in the non-fasted group were not anesthetized and had free access to food and water.

### 2.4. Anesthetic Protocol

Eight mares in this study underwent general anesthesia after a fasting period of 8 h. Pre-anesthesia consisted of detomidine (20 µg/kg IV) followed by methadone (0.1 mg/kg IV). After 10 min, anesthetic induction was achieved with 10% ketamine (2 mg/kg IV) and midazolam (0.1 mg/kg IV) combined in the same syringe. Anesthesia was maintained with isoflurane in 100% oxygen for 90 min. Controlled ventilation was used to maintain ETCO2 at 35–45 mmHg, ventilator pressure at 10 cmH_2_O, positive end-expiratory pressure (PEEP) of 5 cmH_2_O, and a standardized inspiration/expiration ratio of 1:2. The oxygen flow was adjusted to 1 L/100 kg and isoflurane vaporization maintained at 1.5 minimum alveolar concentration (MAC). Mean arterial pressure was kept above 60–70 mmHg using dobutamine infusion (5 µg/kg IV).

### 2.5. Gastric Fluid Sample Collection and Analysis

After clinical examination, all mares had a size 11 nasogastric tube passed through the nasal cavity and down through the esophagus into the stomach. A narrower tube was then passed through the nasogastric tube for gastric juice collection. Both tubes were marked in order to standardize the length of the tube inserted and hence the collection site. A 60 mL syringe was used for aspiration. Next, the nasogastric tube was lowered, and gastric fluid was collected by gravity. The first aliquot (5 mL) was discarded and a 10 mL sample was collected for immediate pH determination (T0).

In the anesthetized group, the transanesthetic period (90 min) was divided into six 15 min sampling intervals (T15–T90). Immediately after recovery from anesthesia (45 ± 1.86 min), gastric juice samples were collected every hour for 24 h (T1 to T24). During this 24 h period, mares were kept in individual stalls and fed concentrate (1% of body weight) twice daily (T4 and T16). Bermuda grass hay and water were offered ad libitum.

Control group mares were sampled after 8 h of fasting, at T0, T15, T30, T45, T60, T75, and T90. No food or water was offered during this period. After 45 min, gastric juice samples were collected every hour for four hours (T1 to T4). During this four-hour period, mares were kept in individual stalls and offered Bermuda grass hay and water ad libitum.

Mares in the non-fasted group had free access to Bermuda grass hay and water. These mares were housed in individual stalls and sampled (T0) 30 min after receiving concentrate (1% of body weight).

Measurements of pH in the first 24 h post-anesthesia, when mares were receiving concentrate, hay, and water, were aimed at determining the impact of eating and/or drinking on gastric juice pH. 

Gastric pH was measured using a digital pH meter (PG1800, Gehaka, São Paulo, Brazil). pH meter calibration in three buffer solutions (neutral/pH 6 to 8, acidic/pH 2 to 6, and alkaline/pH 8 to 20) was carried out to obtain a calibration certificate specificity higher than 95%. After calibration, the pH meter electrode was immersed in gastric juice immediately after sample collection for accurate pH measurement. 

### 2.6. Serum Gastrin Measurement

Whole blood samples (8–10 mL) were obtained from anesthetized group mares. Following skin disinfection, samples were collected by jugular venipuncture into clot activator tubes (BD Vacutainer Systems, Plymouth, UK) and centrifuged for 15 min at 1500× *g* for serum separation. Serum samples were stored in 2 mL tubes (Eppendorf, São Paulo, Brazil) at −3 °C until use. Samples were collected at different time points. The first sample was collected before the anesthetic procedure and used to determine basal gastrin concentrations. The second sample was collected in the recovery room following 90 min of general anesthesia after the mares had been placed in lateral recumbency. The last sample was collected 4 months after anesthesia, 90 min after morning feeding, and without prior fasting. Serum gastrin concentrations were measured using a commercial double-antibody radioimmunoassay (RIA) designed for human use, at an external commercial laboratory (Diagnostics Clinical Analysis, DAC, Pirassununga, SP, Brazil), using a previously validated method of measuring gastrin in equine serum [15]. The test was performed to commercial standards and internal calibration.

### 2.7. Statistical Analysis

Experimental data distribution was tested for normality using the Shapiro–Wilk test. Groups (i.e., anesthetized, control, and not fasted) were compared using Student’s t or the Wilcoxon test, depending on data distribution. Differences in gastrin levels were estimated using ANOVA and the Tukey test. Statistical analyses were performed using Stata software v14. The level of significance was set at 95%.

## 3. Results

Twenty-one mares tolerated nasogastric intubation well and were able to eat hay and concentrate and drink water with the tube in place without difficulties during the 24 h sample collection period. However, in one mare the tube had to be removed after 11 h due to evident discomfort.

### 3.1. Gastric pH

During general anesthesia, the gastric pH remained within the physiological range (4.52 ± 1.69). No significant changes in pH were detected during general anesthesia and dorsal recumbency (*p*  <  0.05). In the control group, gastric pH ranged from weakly acidic to neutral or weakly alkaline during the fasting period (T0 to T90) and did not differ significantly between time points. Gastric pH was lower in anesthetized than in non-anesthetized mares ([4.52 ± 1.69 and 7.04 ± 1.06] respectively; *p*  < 0.05) from T15 to T90. Mean gastric pH values did not differ between anesthetized and control group mares at T0, after 8 h of fasting (Figure 1).

Gastric pH remained alkaline throughout the 24 h post-anesthesia sampling period (Figure 2). Mean pH values above the physiological range for the species were detected, with significant differences (*p* < 0.05) relative to T0 at T5, T8, T9, T11, T13, and T17. Mean gastric pH values did not differ between the control and the anesthetized group from T1 to T4. Gastric pH was weakly alkaline in both groups during this period (Figure 3).

Mares in the non-fasted group had alkaline gastric pH at T0 (7.67 ± 0.13). Mean pH values were higher in these mares than in the control group mares at T0 (6.6 ± 0.63) and in anesthetized mares during general anesthesia (T0 to T90) (Table 1; Figure 4).

In the 24 h post-anesthesia, mean gastric pH was 6.26 ± 1.28 during concentrate intake, 6.71 ± 1.38 during hay intake, and 6.58 ± 1.17 during water intake. Data analysis revealed that water and hay intake had no impact on gastric pH values (*p* < 0.05). In contrast, concentrate intake was associated with lower gastric pH values (*p* = 0.01).

### 3.2. Serum Gastrin Measurement

Analysis of serum gastrin concentrations measured in the fasted state (15.15 ± 3.82), after general anesthesia (20.15 ± 7.65) and 90 min after feeding (19.28 ± 6.26) revealed significant differences between baseline and post-anesthesia values (*p* = 0.03). Serum gastrin concentrations did not differ significantly between the feeding and the post-anesthesia or fasting period (*p* > 0.05) (Figure 5).

## 4. Discussion

Gastric acidity is determined by the pH of gastric secretions and/or contents [16]. Measurements of gastric acidity have been used for many years to characterize peptic disorders and to evaluate response to therapy in human patients [17,18]. In those studies, gastric fluid aspirated from affected patients was typically used to determine acid output and/or pH. In human medicine, intragastric pH monitoring is the gold standard for the detection of changes in gastric physiology and diagnosis of gastrointestinal diseases such as GERD, DGER, and gastritis [8,9].

Prevalence estimates of equine gastric ulcer syndrome (EGUS) in adult horses have been reported to range from 60 to 90 percent, depending on age, performance, and evaluated populations. The disease has significant economic impacts due to reduced athletic performance and treatment costs. Several etiological factors have been implicated in equine gastric ulceration, including fasting, gastric acid clearance (gastric motility and emptying), aggressive gastric juice mucosal factors (hydrochloric acid, pepsin, bile acids, and organic acids), and the epithelial desquamation [4,19].

In equine medicine, research addressing the normal intragastric environment is mostly limited to the fields of nutrition, pharmacology, and sports medicine [1,20,21]. Studies investigating general inhalation anesthesia and dorsal recumbency as risk factors for EGUS development are lacking. Recumbency has been shown to reduce gastric fluid pH in foals. This finding suggests recumbency may increase exposure of the squamous mucosa to gastric acid [1,4]. Increased intra-abdominal pressure [22,23,24] and sphincter relaxation induced by anesthesia [25,26] may also predispose to duodenogastric reflux, increasing the exposure of the gastric mucosa to bile salts [19,27] and predisposing to EGUS development.

The equine stomach secretes variable amounts of hydrochloric acid throughout the day and night, regardless of the presence of feed material. In adult horses, the stomach secretes approximately 1.5 L of gastric juice hourly and acid output ranges from 4 to 60 mmoles hydrochloric acid per hour. The pH of gastric contents ranges from 1.5 to 7.0, depending on the region measured. A near-neutral pH can be found in the dorsal portion of the esophageal region (saccus cecus) near the lower esophageal sphincter, whereas more acidic pHs can be found near the margo plicatus (3.0–6.0) and in the glandular region near the pylorus (1.5–4.0) [28,29,30]. Even though lower than the mean gastric pH in the control group, mean gastric pH values measured from T0 to T90 in this study were within the physiological range for the equine species (i.e., general inhalation anesthesia and dorsal recumbency for 90 min did not affect the gastric environment). This finding suggests preventive pharmacological measures are not required in adult horses submitted to general anesthesia and dorsal recumbency.

Conversely, the mean gastric pH values measured 24 h after recovery from anesthesia (T1 to T24), when mares had free access to food and water, were higher than the physiological limit for the species (2.4 to 5.4) [18]. Higher pH relative to physiological values was also observed after 9.5 h fasting (8 h of fasting plus 90 min of sample collection; mean gastric pH 7.05 ± 1.06) and in the fed state (mean gastric pH 7.38 ± 0.28) in the control group. Similar results were obtained in non-fasted mares (mean gastric pH 7.67 ± 0.43). Post-anesthetic water and/or hay intake had no impact on gastric pH. On the contrary, feed intake promoted gastric acidification, a physiological event resulting from changes in saliva production [20]. Given the copious amounts of non-parietal secretions in the absence of food, alkaline pH is expected in the stomach of fasted horses. Secretion of gastric protectants in response to the constant presence of a nasogastric tube may also have contributed to the elevation of gastric pH in this study [31,32]. However, gastric pH measured at baseline (T0) was also above the physiological range in non-fasted mares and mares submitted to 8 h of fasting.

Studies published to date involving the measurement of gastric pH were carried out in horses kept off feed for several hours. Therefore, the findings of such studies are limited to basal or chemically induced acid secretion [18,33]. Murray and Schusser (1993) [18] demonstrated that equine unfed gastric pH remained highly acidic throughout the 24 h recordings and that there were no circadian differences in gastric acidity (pH). Conversely, in this study, gastric pH was not highly acidic after 8 h of fasting and ranged from neutral to weakly alkaline throughout the 24 h recordings. Mean gastric juice pH ranged from weakly acidic to neutral or weakly alkaline in all groups, regardless of food and water intake (i.e., in the fasted, non-fasted, and fed states).

Under normal conditions, the squamous epithelial mucosa of the equine stomach does not seem to be exposed to a highly acidic pH, as shown by pH measurements in the control and the non-fasted group, and in the 24 h after anesthesia in the anesthetized group. In a study with 27 adult horses, the average surface pH of the squamous epithelial mucosa in the dorsal portion of the stomach was 5.5 and the pH of the squamous epithelial mucosa adjacent to the margo plicatus was 4.1 [34]. Lower gastric acidity in response to food intake is thought to result from the buffering effect of salivary bicarbonate and the absorption of gastric acids by food materials. Continuous food intake may sufficiently neutralize gastric acidity [35]. However, 8 h of food deprivation did not decrease gastric pH to a high acid pH, as evidenced in this study. In contrast, lower gastric pH values were observed in mares during general anesthesia and dorsal recumbency, probably due to continuous acid secretion in the absence of feed [18,20].

Gastrin levels reflect enhanced gastric stimulation and the production of acids in the stomach. Therefore, measurement of gastrin concentration is often used to diagnose gastric disorders in equine clinical practice. In this study, serum gastrin concentrations did not differ significantly between the fasting and feeding periods, with values ranging from 15.15 ± 3.82 pg/ml during fasting to 19.28 ± 6.26 pg/mL after feeding. Similar values have been reported in normal horses during fasting (less than 8 pg/mL to 17.5 pg/mL) [36,37,38,39] and in the fed state (21.1 ± 15.6 pg/mL) [40].

In this study, mean gastrin levels measured after general inhalation anesthesia and dorsal recumbency were higher (20.15 ± 7.65 pg/mL) than baseline. This finding may be explained by the increased intra-abdominal pressure induced by dorsal recumbency. Higher pressure exerted on the stomach stimulates gastrin production by G cells in the gastric mucosa [37]. Gastric distension caused by duodenogastric reflux and the passage of alkaline duodenal fluid into the stomach are the primary factors responsible for enhanced gastrin secretion. Increased gastrin concentrations (69.0 ± 32.2 pg/mL) have been reported in horses with gastric distension caused by acute duodenal reflux secondary to strangulating obstruction of the small intestine [40]. Higher gastrin levels observed during the immediate post-anesthetic period may also have reflected the inhibition of gastrin-induced histamine secretion and acid response, possibly due to the suppressive effects of anesthetic agents on parietal and enterochromaffin-like cells. However, the resultant increase in serum gastrin is not sufficient to lower the gastric pH after anesthesia [38,39,41].

The effects of general anesthesia in dorsal recumbency and food intake on gastrin production by G cells in the stomach were also compared. Blood samples were collected 4 months after the anesthetic procedure, and 90 min after the morning meal. Different from observations made during recovery from anesthesia, feeding did not increase serum gastrin levels (compared to the fasting period). Since horses consume small amounts of food throughout the day, gastrin production increases gradually and reaches peak levels late [42,43,44]. The findings of this study suggest elevated gastrin levels measured during recovery from anesthesia result from general anesthesia and dorsal recumbency and are unrelated to fasting.

One limitation of the present study was the lack of a gastroscopic assessment for continuous evaluation of the stomach, gastric juice, and nasogastric tube position. Another limitation of the present study was the small number of mares used in each group, a fact that increases individual variation.

## 5. Conclusions

General inhalation anesthesia and dorsal recumbency do not affect gastric juice pH, which remains acidic and within the physiological range during the procedure. Gastric juice pH was alkaline after recovery from anesthesia, after 8 h of fasting without anesthesia and recumbency, and during food and water intake. Serum gastrin levels increase in response to general inhalation anesthesia in dorsal recumbency and are not influenced by fasting. Preventive pharmacological measures possibly are not required in horses submitted to general anesthesia and dorsal recumbency.

## Figures and Tables

**Figure 1 animals-14-01183-f001:**
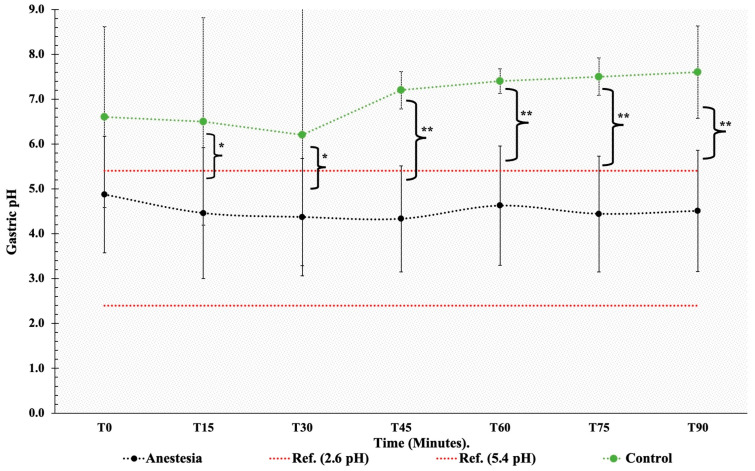
Mean gastric pH from the control group (--∙--) and anesthetized group (--∙--) during general anesthesia, T0 to T90. T sampling time in minutes. Red lines represent physiological values. * = indicates a significant difference (*p* < 0.05). ** = indicates a significant difference (*p* < 0.01).

**Figure 2 animals-14-01183-f002:**
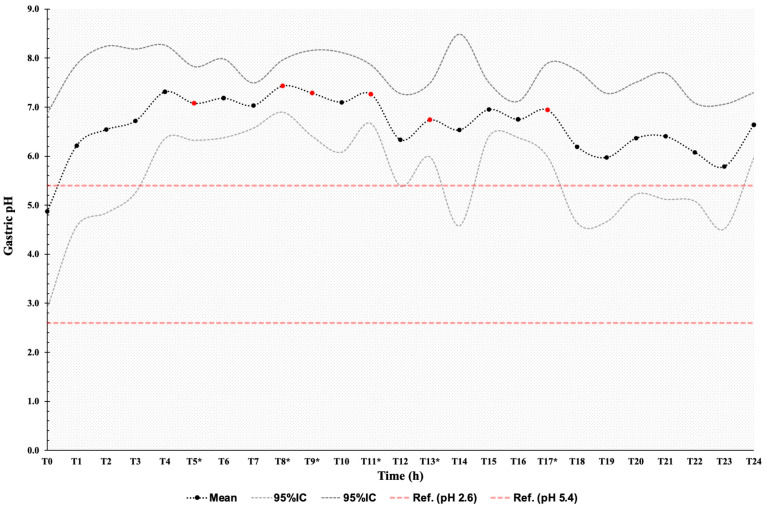
Mean gastric pH from the anesthetized group after general anesthesia with free access to water, hay, and concentrate during the 24 h period. T: sampling time in hours. * and • significant difference between times (*p* ≤ 0.05). The gray line represents the 95% confidence interval limits.

**Figure 3 animals-14-01183-f003:**
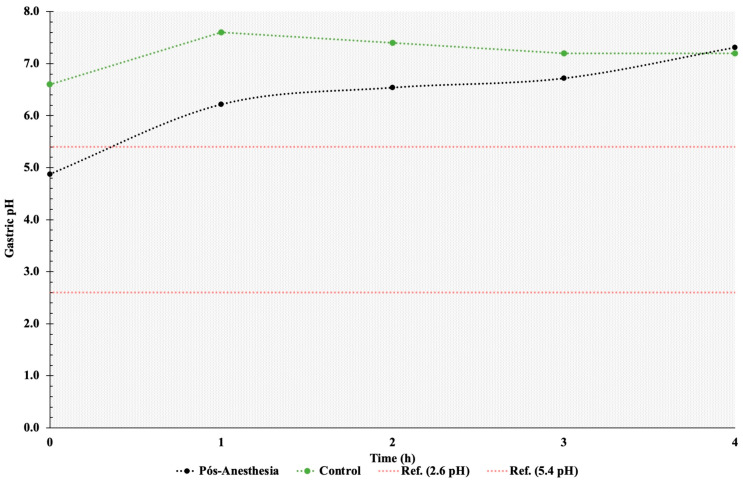
Comparison of mean gastric pH from the control group (--∙--) and anesthetized group (--∙--) during the first four hours after general anesthesia (T1 to T4), with free access to food and water. T represents the sampling time in hours. Red lines represent physiological values.

**Figure 4 animals-14-01183-f004:**
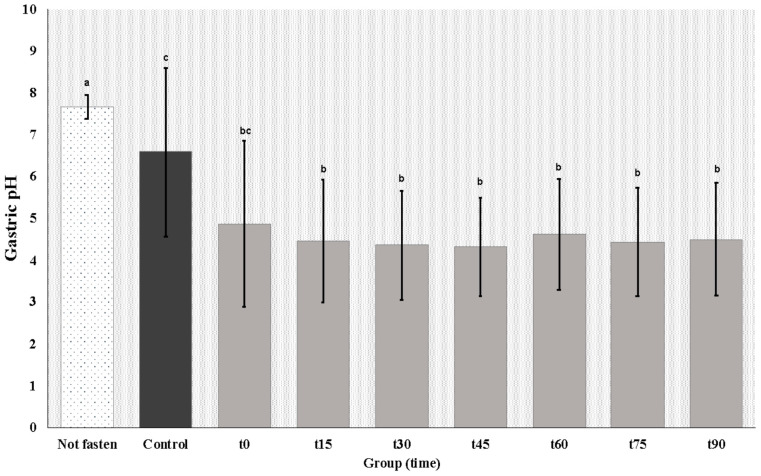
Mean gastric pH from the non-fasted group (T0), control group after 8 h fast (T0), and anesthetized group during anesthesia (T0 to T90). ^a,b,c^ uncommon superscript letters differ significantly (*p* < 0.05).

**Figure 5 animals-14-01183-f005:**
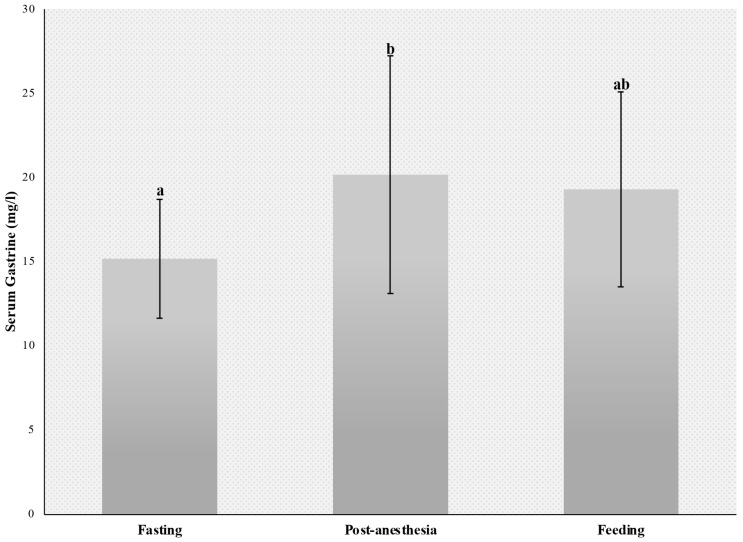
Mean serum gastrin concentrations after fasting, post-anesthesia, and 90 min after feeding. ^a,b^ uncommon superscript letters differ significantly (*p* < 0.03).

**Table 1 animals-14-01183-t001:** Mean gastric juice pH in the control group (T0), non-fasted group (T0), and anesthetized group (T0 to T90).

Group and Time	pH	*p*-Value (vs. Not Fasten)	*p*-Value (vs. Control)
Non-fasted T0	7.67 ± 0.13 ^a^	Ref.	0.0119
Control T0 *	6.6 ± 0.63 ^c^	0.0119	Ref.
Anaesthetized T0 *	4.88 ± 0.84 ^bc^	0.0006	0.1056
T15	4.46 ± 0.62 ^b^	<0.0001	0.028
T30	4.37 ± 0.55 ^b^	<0.0001	0.0168
T45	4.33 ± 0.50 ^b^	<0.0001	0.0111
T60	4.63 ± 0.56 ^b^	<0.0001	0.0286
T75	4.44 ± 0.55 ^b^	<0.0001	0.0186
T90	4.51 ± 0.57 ^b^	<0.0001	0.0244

* After 8 h fast. ^a,b,c^ uncommon superscript letters differ significantly (*p* < 0.05).

## Data Availability

Data supporting reported results can be found at http://dedalus.usp.br, accessed on 15 March 2024 and https://repositorio.usp.br/index.php, accessed on 15 March 2024.

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
