# Peer review of "Evaluation of Gastric pH and Gastrin Concentrations in Horses Subjected to General Inhalation Anesthesia in Dorsal Recumbency"

_animals, 2024, doi:10.3390/ani14081183_

Round 1

Reviewer 1 Report

Comments and Suggestions for Authors

The article titled Evaluation of Changes in Gastric pH and Gastrin Concentrations in Horses Subjected to General Inhalation Anesthesia in Dorsal Recumbency addresses an important topic, namely gastric problems in horses following anesthesia. While the work is intriguing, certain aspects need correction before publication:

Simple summary and abstract: The authors include gastritis as a frequent pathology when it is not. Only equine gastric ulcer syndrome should be cited as a highly prevalent disease in horses. Gastritis in adult horses has not been demonstrated without an inflammatory component external to the stomach (enteritis, IBD, etc.), unlike in other species such as dogs.

2.3. Experimental Groups. Prior gastric problems are not considered. It is necessary to perform a prior gastroscopy to ensure that mares did not suffer from EGUS before the study (which could bias the results).

2.4. Anesthetic Protocol. Were any other medications administered to the mares before, during, or after anesthesia (such as NSAIDs or antibiotics)? Were the mares anesthetized solely for the study without any other intervention during the anesthesia period? If so, it should be specified.

2.6. Serum Gastrin Measurement. Were the samples taken from the jugular vein after skin disinfection? Why were they not taken from the catheter? I understand that a catheter was used to administer premedication and maintain a horse under general anesthesia.

5. Conclusions. Preventive pharmacological measures are not required in horses submitted to general anesthesia and dorsal recumbency. This sentence should be removed from the conclusions of the study. It is unknown whether the mares suffered from gastric ulcers due to contact with hydrochloric acid in the squamous region when placed in dorsal recumbency. It is necessary to perform a gastroscopy to ensure that preventive measures are not necessary.

Author Response

Dear Reviewer,

We are enclosing herewith the revised manuscript entitled “Evaluation in gastric pH and gastrin concentrations in horses subjected to inhaled general anesthesia in dorsal recumbency”.

The manuscript was rewriten according to the reviewers comments and suggestions. We took care of all the comments and suggestions that the reviewers had done carefully and we made all the changes suggested and respond all the comments individually. So we hope the preparation of this revision will permit a positive final decision on our report's suitability for publication in Animals.

Thank you very much for the additional comments.

Best regards.

Doria RGS et al. 

Reviewer 1

Comments and Suggestions for Authors

Simple summary and abstract: The authors include gastritis as a frequent pathology when it is not. Only equine gastric ulcer syndrome should be cited as a highly prevalent disease in horses. Gastritis in adult horses has not been demonstrated without an inflammatory component external to the stomach (enteritis, IBD, etc.), unlike in other species such as dogs.

Answer: The text was corrected, as suggested.

2.3. Experimental Groups. Prior gastric problems are not considered. It is necessary to perform a prior gastroscopy to ensure that mares did not suffer from EGUS before the study (which could bias the results).

Answer: The mares remained housed in the University Equine Facility for at least a year, released on pasture, fed with Coast cross pasture, Coast cross hay, commercial feed, mineral salt and water ad libitum. None of the animals showed any illness, especially signs of colic, during this period. These are animals that lived together, free, dewormed every 6 months, vaccinated according to the vivarium's vaccination protocol. Although gastroscopy was a limitation of the study, the condition in which they lived reduced the chance of gastric ulceration and no clinical, hematological or biochemical changes were observed in pre-experimental examinations.

2.4. Anesthetic Protocol. Were any other medications administered to the mares before, during, or after anesthesia (such as NSAIDs or antibiotics)? Were the mares anesthetized solely for the study without any other intervention during the anesthesia period? If so, it should be specified.

Answer: No medications were administered to the mares before, during, or after anesthesia. The mares were anesthetized during 90 minutes solely for the study without any other intervention during the anesthesia period.  

2.6. Serum Gastrin Measurement. Were the samples taken from the jugular vein after skin disinfection? Why were they not taken from the catheter? I understand that a catheter was used to administer premedication and maintain a horse under general anesthesia.

Answer: The samples were taken from the jugular vein to standardize with the last sample, collected 4 months after anesthesia. At this time point, the mares were not using catheter.

  1. Conclusions. Preventive pharmacological measures are not required in horses submitted to general anesthesia and dorsal recumbency. This sentence should be removed from the conclusions of the study. It is unknown whether the mares suffered from gastric ulcers due to contact with hydrochloric acid in the squamous region when placed in dorsal recumbency. It is necessary to perform a gastroscopy to ensure that preventive measures are not necessary.

Answer: The sentence was modified, as suggested.

Reviewer 2 Report

Comments and Suggestions for Authors

Congratulations on producing excellent research and a flawless manuscript. The research question was well defined and is likely one that equine surgeons are concerned about, given the incidence of gastric ulcers and the abnormal positioning of a horse during surgery in dorsal recumbency.

It is refreshing to read a report where authors are confident in their results and are brave enough to state there is no problem. In addition, they recommend that no intervention is necessary. A well-designed study with attention to including relevant control groups.

Author Response

Dear Reviewer,

We are enclosing herewith the revised manuscript entitled “Evaluation in gastric pH and gastrin concentrations in horses subjected to inhaled general anesthesia in dorsal recumbency”.

The manuscript was rewriten according to the reviewers comments and suggestions. We took care of all the comments and suggestions that the reviewers had done carefully and we made all the changes suggested and respond all the comments individually. So we hope the preparation of this revision will permit a positive final decision on our report's suitability for publication in Animals.

Thank you very much for the additional comments.

Best regards.

Doria RGS et al. 

Reviewer 2

Congratulations on producing excellent research and a flawless manuscript. The research question was well defined and is likely one that equine surgeons are concerned about, given the incidence of gastric ulcers and the abnormal positioning of a horse during surgery in dorsal recumbency.

It is refreshing to read a report where authors are confident in their results and are brave enough to state there is no problem. In addition, they recommend that no intervention is necessary. A well-designed study with attention to including relevant control groups.

Answer: Thank you very much for the special attention. We are very grateful for the positive comments on the survey.

Reviewer 3 Report

Comments and Suggestions for Authors

The manuscript "Evaluation of Changes in Gastric pH and Gastrin Concentrations in Horses Subjected to General Inhalation Anesthesia in Dorsal Recumbency" focus an original interesting subject regarding horse’s health, since gastric ulcers are common in this species and are associated to stress. Since general anaesthesia causes stress and fasting is also implicated in equine gastric ulcer syndrome, this study is pertinent and could help practitioners how to best manage these situations.

Tittle: The tittle "Evaluation of Changes in Gastric pH and Gastrin Concentrations in Horses Subjected to General Inhalation Anesthesia in Dorsal Recumbency" is adequate, but in my opinion could be only: Evaluation in Gastric pH and Gastrin Concentrations in Horses Subjected to General Inhalation Anesthesia in Dorsal Recumbency.

Simple summary is correct.

Line 17: “…evaluates the pH of the stomach of anesthetized horses lying in the dorsal recumbency and also…” Correct to: …evaluates the pH of the stomach of anesthetized horses in dorsal recumbency and also…

The Abstract is correct and elucidates the content of the manuscript.

Lines 30 “… then at 15 min intervals (T15 - T90).” Correct to: …then at 15 min intervals during anesthesia (T15 - T90).

The introduction section is satisfactory.

Line 60- “Foals and athletic horses…” I suggest to remove foals from the sentence. In foals the aetiology differs from adult horses, being related to hypoxia, etc…

The materials and methods seem adequate.

Line 163-164– “a commercial double-antibody radioimmunoassay designed for human use (Diagnostics Clinical Analysis, DAC, Pirassununga, SP, Brazil).” Can the authors give information if this commercial radioimmunoassay is validated for horses? If so, add reference.

The results are sound.

Lines 257- “Mean gastric pH values measured from T0 to T90 in this…”. I Suggest that this sentence begins with: Even though lower than mean gastric pH in control group, mean gastric…

The discussion is correct.

Lines 326-327: Authors should be more conscious regarding limitations of the study. The small number of mares in each group is a limitation…

Author Response

Dear Reviewer,

We are enclosing herewith the revised manuscript entitled “Evaluation in gastric pH and gastrin concentrations in horses subjected to inhaled general anesthesia in dorsal recumbency”.

The manuscript was rewriten according to the reviewers comments and suggestions. We took care of all the comments and suggestions that the reviewers had done carefully and we made all the changes suggested and respond all the comments individually. So we hope the preparation of this revision will permit a positive final decision on our report's suitability for publication in Animals.

Thank you very much for the additional comments.

Best regards.

Doria RGS et al. 

Reviewer 3

 Tittle: The tittle "Evaluation of Changes in Gastric pH and Gastrin Concentrations in Horses Subjected to General Inhalation Anesthesia in Dorsal Recumbency" is adequate, but in my opinion could be only: Evaluation in Gastric pH and Gastrin Concentrations in Horses Subjected to General Inhalation Anesthesia in Dorsal Recumbency.

Answer: The tittle was modified, as suggested.

Simple summary is correct.

Line 17: “…evaluates the pH of the stomach of anesthetized horses lying in the dorsal recumbency and also…” Correct to: …evaluates the pH of the stomach of anesthetized horses in dorsal recumbency and also…

The Abstract is correct and elucidates the content of the manuscript.

Lines 30 “… then at 15 min intervals (T15 - T90).” Correct to: …then at 15 min intervals during anesthesia (T15 - T90).

The introduction section is satisfactory.

Line 60- “Foals and athletic horses…” I suggest to remove foals from the sentence. In foals the aetiology differs from adult horses, being related to hypoxia, etc…

The materials and methods seem adequate.

Line 163-164– “a commercial double-antibody radioimmunoassay designed for human use (Diagnostics Clinical Analysis, DAC, Pirassununga, SP, Brazil).” Can the authors give information if this commercial radioimmunoassay is validated for horses? If so, add reference.

The results are sound.

Lines 257- “Mean gastric pH values measured from T0 to T90 in this…”. I Suggest that this sentence begins with: Even though lower than mean gastric pH in control group, mean gastric…

The discussion is correct.

Lines 326-327: Authors should be more conscious regarding limitations of the study. The small number of mares in each group is a limitation…

Answer: All suggestions were accepted, modified and highlighted in the text.

Round 2

Reviewer 1 Report

Comments and Suggestions for Authors

Thank you for correcting and addressing the comments. I believe the article has improved sufficiently to warrant publication.